# Aggressive Pyogenic Spondylitis Caused by *S. constellatus*: A Case Report

**DOI:** 10.3390/diagnostics12112686

**Published:** 2022-11-04

**Authors:** Nenad Koruga, Alen Rončević, Anamarija Soldo Koruga, Dario Sabadi, Domagoj Drenjančević, Ana Prica, Tatjana Rotim, Tajana Turk, Domagoj Kretić

**Affiliations:** 1Department of Neurosurgery, University Hospital Center Osijek, 31000 Osijek, Croatia; 2Faculty of Medicine, Josip Juraj Strossmayer University of Osijek, 31000 Osijek, Croatia; 3Department of Neurology, University Hospital Center Osijek, 31000 Osijek, Croatia; 4Department of Infectious Diseases, University Hospital Center Osijek, 31000 Osijek, Croatia; 5Department of Microbiology, Parasitology and Clinical Laboratory Diagnostics, Faculty of Medicine, Josip Juraj Strossmayer University of Osijek, 31000 Osijek, Croatia; 6Department of Diagnostic and Interventional Radiology, University Hospital Center Osijek, 31000 Osijek, Croatia

**Keywords:** *Streptococcus constellatus*, spondyilitis, epidural abscess, porencephaly, ampicillin

## Abstract

*Streptococcus constellatus* (SC) is a species of *Streptococcus* belonging to the *Streptococcus anginosus* group, along with *Streptococcus anginosus* and *Streptococcus intermedius*. Despite its commensal nature, underlying risk factors and medical conditions might lead to various anatomic site infections caused by this opportunistic pathogen. Although SC infections have mostly been associated with bacteremia, some case reports of abscess and empyema formation have been documented. Herein, we report a case of a middle-aged female patient who initially presented with radiculopathy symptoms. Subsequent neurologic imaging revealed a pyogenic abscess along paravertebral muscles, which was found to be caused by SC. The patient was successfully treated with abscess drainage from the lumbar zone and antibiotics, and the symptoms of radiculopathy have completely resolved.

## 1. Introduction

### 1.1. Streptococcus Anginosus Group

*Streptococcus* is a genus of Gram-positive spherical-shaped bacteria belonging to the family of *Streptococcaceae*. The majority of species that belong to this genus are oxidase-negative and catalase-negative, as well as facultative anaerobes [1]. The total number of identified species in the genus *Streptococcus* already exceeds 100, and with technological advancements and the accessibility of next-generation sequencing technologies, this number is likely to increase even more.

*Streptococcus anginosus* group (SAG) consists of three species of streptococci—*Streptococcus anginosus* (SA), *Streptococcus intermedius*, and *Streptococcus constellatus* (SC). This group of streptococci was formerly known as the *milleri* group though this nomenclature has generally been abandoned. These commensal organisms have been isolated from various anatomical sites and have been shown to cause pyogenic infections that lead to the formation of abscesses [2]. The SAG species are generally isolated from intra-oral samples and are known for their association with dental plaques and periodontal pathologies. Advanced sequencing tools, such as rRNA sequence analysis, have verified the phylogenetic relatedness of SA, *S. intermedius*, and SC [3].

The most widely studied SAG is SA. Even though it is a part of the normal flora of the human body, this opportunistic pathogen can cause serious infections in immunocompromised patients. The most common infections it causes are brain, lung, and liver abscesses; however, other clinical presentations have also been described [4]. Bacteremia caused by SA is not as common, and an abscess is frequently the main source [5]. According to the results presented in this study [5], the most prevalent infections caused by SAG were infections of the skin or soft tissue, followed by intra-abdominal infections, among which liver abscesses were the most common. On the other hand, infections of other tissues such as bone (osteomyelitis), respiratory tract (lung abscesses or empyema), and nervous tissue (brain abscess) were much less represented. Regarding the treatment of infections caused by species from SAG, they usually respond well to penicillin and ampicillin treatments. Significantly, many of the strains are resistant to macrolides and tetracyclines.

### 1.2. Streptococcus Constellatus

As previously stated, SC is a part of SAG and has the potential to spread to physiologically sterile anatomic sites and cause infections, which is predominantly observed in immunodeficient patients. It is a part of the normal mouth, gut, and urogenital tract flora. According to its anatomic distribution, SC may cause suppurative infections of multiple anatomic sites mostly in patients with immunosuppression or undergoing invasive procedures. A retrospective cohort study evaluating the pyogenic potential of species belonging to the SAG reported that SC was mostly associated with bacteremia, while the presence of abscesses or empyema was not as common [6]. Despite that, there have been some case reports of patients presenting with abscesses or empyema caused by SC.

Despite the fact that infections caused by SC are rather uncommon, some risk factors have been identified, such as male gender, smoking, frequent alcohol consumption, oncological conditions, chronic diseases of the respiratory tract (e.g., chronic obstructive pulmonary disease), intra-oral diseases (especially periodontal illness), diabetes mellitus, hepatitis, and HIV infections [7]. Furthermore, there are several means of SC dissemination to various anatomic sites where it can cause the infection: aspiration of intra-oral secretion, direct spread to adjacent tissues or due to trauma or surgical interventions, and hematogenous spread [8]. The most common microbiological tool for identifying SC is culture, although multiplex polymerase chain reaction (PCR) has also been used. Specific antibiotic treatments and their duration of SC infections differed in case reports, while the most commonly administered antibiotics were β-lactam/β-lactam inhibitor combination, carbapenem, clindamycin, ceftriaxone, metronidazole, and ciprofloxacin [7].

Here, we present a rare case of a female patient presenting with pyogenic spondylitis caused by SC with concomitant infection along the right-sided paravertebral and paracostal musculature.

## 2. Case Report

A 43-year-old female patient was admitted to the Department of Neurosurgery, University Hospital Center, Osijek, due to lumbar pain and L3 radiculopathy. Symptoms were first presented ten days before admission, and the patient underwent symptomatic therapy at a local hospital. Despite therapy, the symptoms persisted, and a computed tomography (CT) scan of the whole spine was obtained on the day of admission at another hospital. The right-sided gas inclusions were presented along the cervical, thoracic, and lumbar paravertebral muscles, with the largest collection at the segments of the L3 and L4 vertebra. Moreover, the purulent spondylitis of the L3 vertebra was noted with concomitant purulent epidural content and gas inclusion (Figure 1). Additionally, a CT scan of the brain revealed pneumocephalus: multiple gas inclusions along the left-sided brain hemisphere and at the skull base (Figure 2). According to her medical history, she had a history of previous COVID-19 infection, and she did not receive the SARS-CoV-2 vaccine. Additionally, she had a history of unspecified cardiomyopathy for which the patient did not provide an explanation, nor were there detailed medical records pertaining to this comorbidity. A physical examination of the patient revealed that the key muscle strength assessment of the right lower limb was 3/5 and for the left lower limb was 4+/5, respectively. No other chronic or familiar diseases were found via interview or from the medical records. The patient was disoriented upon admission, and her Glasgow Coma Scale (GCS) score was 12 (eyes 3, verbal response 4, motor response 5). Based on the clinical findings as well as radiologic imaging, she underwent urgent surgery, which was performed with local anesthesia in a regular fashion. The anesthesiologist was reluctant to perform endotracheal anesthesia due to the intraoral condition of the patient, carious teeth, and the assumed duration of the surgery, which lasted up to ten minutes. Upon skin incision and soft tissue preparation, the fascia was incised, and the subfascial space was opened. The drainage was placed along the right-sided paravertebral muscles at the level of L3 and L4 vertebrae and left in situ for the next three weeks. Despite the clinical condition of our patient, she was not hospitalized in the intensive care unit (ICU).

During the early postoperative period, intravenous metronidazole, vancomycin, and meropenem were administered as an early empirical antibiotic treatment at the doses of 500 mg every 6 h, 1 g every 12 h, and 2 g every 8 h, respectively. Intraoperatively, the sample of purulent content was obtained for microbiological testing, which later revealed the presence of SC, identified by Bruker MALDI Biotyper^®^. Based on the microbiological assay of antibiotic sensitivity testing, treatment was modified, and intravenous antibiotic treatment was administered—metronidazole, fluconazole, ampicillin, and cefepime, while previously administered empirical antibiotics were discontinued. The obtained hemoculture was also microbiologically tested and was found sterile. The patient underwent an echocardiogram with uneventful findings. The left ventricle was of normal size with an ejection fraction of 61%. No clear signs of pericardial effusion or endocarditis were detected with this diagnostic method.

Laboratory testing at the admission pointed to the normal level of leukocytes, increased level of C-reactive protein (CRP; 126.5 mg/L), and mildly increased levels of aspartate aminotransferase (AST), alanine aminotransferase (ALT) and gamma-glutamyl transferase (GGT) (Table 1). Consistent laboratory examinations and antibiotic administration were applied and evaluated throughout the whole hospital stay at our department which lasted for a total of thirty days. Due to the highly increased values of D-dimers, low molecular weight heparin at the dose of 40 mg was administered twice per day. On the eighteenth postoperative day, laboratory examination revealed mildly higher values of leukocytes (10,9), decreased CRP values (29.8 mg/L), normal values of procalcitonin, and decreased values of D-dimers. Based on improvements in the clinical status of the patient and laboratory tests, the only antibiotic treatment for the remainder of the stay was ampicillin at the dose of 3 g every 4 h. The patient underwent a Color Doppler Ultrasound of the lower limbs, which excluded the presence of thrombosis, and an ultrasound of the abdomen, which excluded any pathologic findings. On the day of the discharge, laboratory testing revealed normal values of leukocytes, higher levels of liver enzymes, and decreased levels of CRP (16.6 mg/L). GCS of the patient upon discharge was 15, the patient was independent and the radiculopathy-related symptom, i.e., the right-sided leg pain was completely resolved.

A follow-up CT scan of the brain revealed the complete resorption of previously described gas inclusions. The follow-up magnetic resonance imaging (MRI) of the whole spine revealed encapsulated purulent content along the thoracic medulla, cauda equina, and hyperintense signal on the T2-weighted image of the right-sided pedicle of the L3 vertebra (Figure 3). A follow-up laboratory examination one month after discharge revealed completely normal findings.

## 3. Discussion

As concluded by Kobo et al. [6], SC infections are usually presented as bacteremia, whereas abscess or empyema formation is infrequent. Pyogenic infections caused by SC are mostly described as lung infections, i.e., empyema [4,7,9]. Recently, Jiang et al. [4] conducted a retrospective study that revealed that the SC infection is the most prevalent in a middle-aged group of patients with a male predominance of 2:1. It should be noted that our patient was female. Additionally, the same study revealed *S. anginosus* as the most common bacteria from SAG that causes symptomatic infections. In terms of patient age and the onset of infection, patients between 35 and 54 years of age are exposed to the highest risk of infection (around 30%)—as was the case with our patient [4]. Regarding other identified risk factors, our patient suffered from a periodontal disease, which was previously documented to increase the risk of infection by SAG [7]. In addition, she was not a smoker and denied frequent alcohol consumption.

As previously stated, SC is found in various body compartments and is not considered a pathogen per se. Clinically relevant infections caused by SC are in the majority of cases described in patients with weakened immune systems (patients undergoing immunosuppressive therapy or immunocompromised patients due to other medical conditions). Therefore, SC has been reported as a causative organism in a series of diseases, such as liver abscesses, empyema, and mediastinitis [9]. However, there has been some debate about SC being the single causative pathogen in these infections. Some researchers argue that these infections are of multimicrobial etiology.

To the best of our knowledge, and based on the review of the published literature, similar cases of pyogenic paracostal and paravertebral abscesses were not yet described. There are only a handful of case reports reporting pyogenic spondylitis caused by SC [10,11,12,13,14]. The case that was similar the most in the published literature was by Jin and Yin in which they presented a patient diagnosed with pyogenic spondylitis caused by SC [10]. That patient was of similar age as the patient in our case report; however, he was male and obese, with a body mass index (BMI) of over 40 kg/m^2^. Similar to the patient we described, the clinical presentation was signified by paraspinal tenderness at the level of L3–L5 vertebrae, as well as reduced muscle strength of lower limbs. Subsequent radiology imaging (MRI and CT scans) had clearly shown abnormal findings in the aforementioned lumbar levels of the spine. In contrast to our case, SC was initially isolated from the blood culture after which the diagnosis of pyogenic spondylitis caused by SC was finally confirmed. After the systemic administration of antibiotics, the patient recovered, but he still suffered from weakened muscle strength and hypoesthesia. Another case that described a patient diagnosed with pyogenic spondylitis was by Gangone et al. [11]. This was an elderly patient (72 years of age) with the main complaint of lower back pain without radiation to the lower extremities. Paraspinal tenderness was also pronounced, though there was no sensory or motor deficit. CT-guided biopsy at the level of lower thoracic vertebrae Th 10–Th 11 was utilized to confirm the diagnosis of spondylodiscitis caused by SC. However, in this case, SC was not the only isolated bacterium—there was a synergistic infection by *Streptococcus viridans*. This patient was treated with intravenous antibiotics and made a full recovery. Lim et al. [12] reported a case of a 14-year-old male teenager with a history of progressively worsening lower back pain for one year, accompanied by a weight loss of more than 10 kg and loss of appetite. However, there was no motor or sensory deficit revealed during the physical examination. Moreover, the patient did not have any of the prior mentioned risk factors (dental disease, surgeries, trauma, etc.). In contrast to the previously mentioned case reports, this patient did not have pronounced paraspinal tenderness over the affected area. MRI of the lumbosacral spine revealed a decrease in the L4–L5 intervertebral disc space and disc destruction with minor paraspinal collection. The diagnostic method that was used was also a CT-guided biopsy and a pure culture of SC was grown from the sample. After 6 weeks of intravenous antibiotic treatment, the patient fully recovered. Furthermore, Wang et al. [13] reported a case of recurrent vertebral osteomyelitis and psoas muscle abscess in a 48-year-old male patient with an atrial septal defect. In this specific case, there were two causative agents, i.e., SC and *Fusobacterium nucleatum*. The patient was successfully treated conservatively with antibiotics and the likely source of infection was dental disease. The case of pyogenic spondylodiscitis due to poor oral health and which was caused by SC was also described by Potsios et al. [14]. This was a 64-year-old immunocompromised male patient presenting with sciatica and back pain and was also effectively treated with antibiotics.

Interestingly, Dai et al. [15] conducted a retrospective study in which they described a series of eleven patients treated for spinal epidural abscesses (SEAs). Of these, only one case was identified as SC infection with concomitant back pain, limb numbness, weakness, and fever. In this study, authors put an emphasis on the importance of thorough clinical and physical examination, as well as accompanying radiology imaging (CT and MRI scans), comprehensive laboratory testing, and microbiological analyses (blood culture and others). This is of crucial importance in patients presenting without any symptoms of the aforementioned clinical triad (focal spinal pain, neurologic deficit, and fever), which were less specific for the group of patients with SEA [15]. Therefore, a proper approach to treatment and correct diagnosis in such patients should be ensured by observing clinical symptoms, laboratory, and radiological findings.

In cases of spondylitis due to SC infection, systemic antibiotic treatment should be considered as the first-choice treatment, and intravenous antibiotics should be administered as soon as possible, although in cases of refractory analgesics and steroid treatment, surgical treatment should be considered. Surgical treatment is considered in cases of severe pain, motor, or sensory deficits due to the radicular or spinal cord compression caused by abscesses, etc., as was the case in our patient.

The main goals of surgical treatment are the decompression of nervous tissue, spinal stabilization based on radiological findings, and the evacuation and drainage of abscess, which allows for proper microbiological assessment and evaluation for further antibiotic treatment [16]. When antibiotic treatment is indicated, it should be administered according to bacterial culture results, although it might be initiated empirically depending on the condition of the patient. Even though many patients report the alleviation of symptoms in less than 6 weeks, antibiotics should be administered for up to eight weeks, according to previously conducted studies [16].

The initial course of our patient’s preoperative symptoms pointed to a possibility of lumbar radiculopathy caused by degenerative spine disease (spinal stenosis), due to which she underwent unsuccessful analgesics treatment. Besides the overall clinical status of our patient, we have to emphasize that her unfavorable health, family, and socio-economic status were evaluated and considered. In addition, we have to emphasize that similar conditions that do not respond to conservative treatments should undergo radiological scanning after which, depending on the findings, a prompt surgical and antibiotic treatment should be strongly considered.

## 4. Conclusions

Lower back pain and radiculopathy that do not respond well to conservative treatment should be further assessed by a physician as soon as possible. The diagnostic work-up should include extensive radiologic evaluation and, if spondylitis is suspected, imaging-guided or open surgery sampling for microbiological analyses should be performed. Acquiring adequate samples is essential in the diagnosis of spondylitis caused by SC. This enables microbiological analyses (cultures) and antibiotic sensitivity tests. Empirical antibiotic treatment prior to microbiological testing should be avoided—it should be reserved for patients with rapid clinical worsening. The surgical approach should be considered a second-tier treatment except in cases with clearly presented and radiographically confirmed pyogenic content in surgically favorable anatomic areas and concomitant symptoms.

## Figures and Tables

**Figure 1 diagnostics-12-02686-f001:**
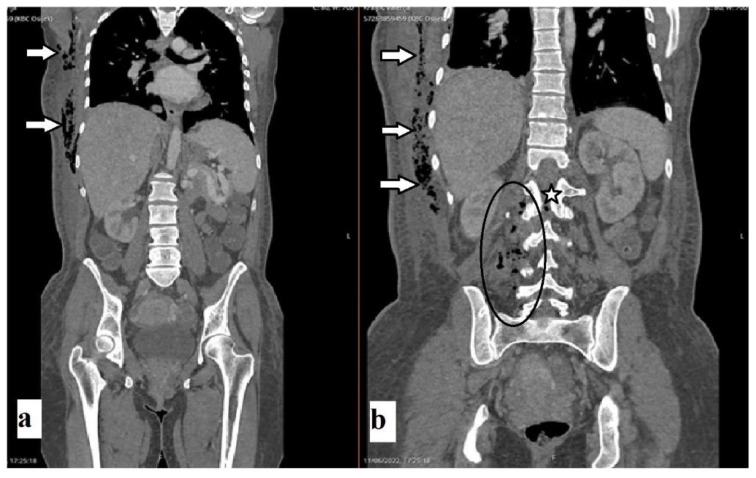
Non-enhanced coronal CT scans of the abdomen, thorax, and spine (**a**,**b**) revealed axillar and paracostal gas inclusions (arrows). Scans revealed paravertebral abscess (**b**) and gas inclusions at the level of L3 (oval) and gas inclusion in the spinal canal (star).

**Figure 2 diagnostics-12-02686-f002:**
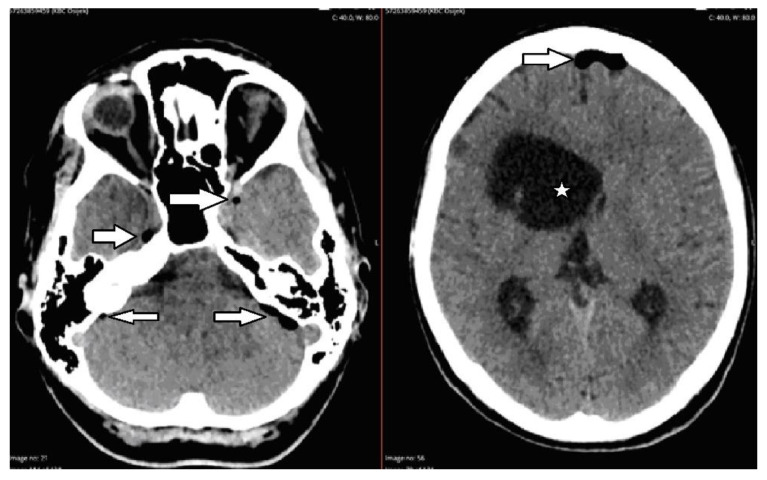
Non-enhanced axial CT images on admission revealed multiple intracranial gas inclusions (arrows). Porencephaly of the right-sided frontal horn of the ventricular system was noted years prior (star).

**Figure 3 diagnostics-12-02686-f003:**
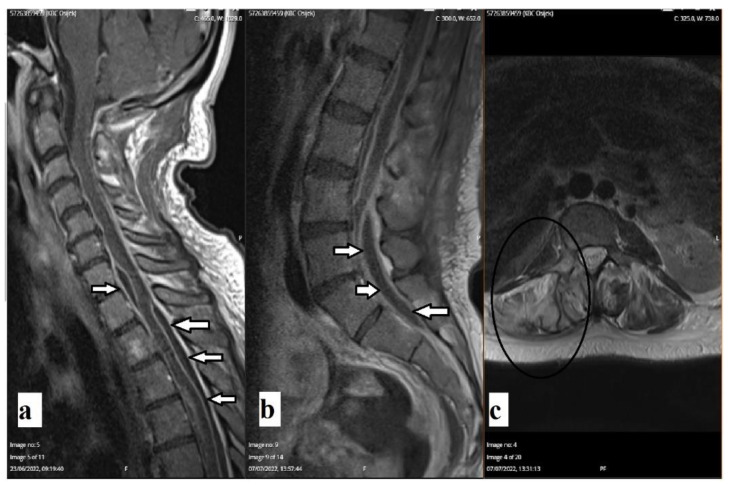
Enhanced sagittal T1-weighted MRI scans of the cervical and thoracic spine (**a**,**b**) revealed encapsulated purulent content along thoracic medulla and cauda equina (arrows). An axial T2-weighted MRI scan (**c**) revealed hyperintensity of the right-sided pedicle of the vertebra L3 and paravertebral soft tissue (oval).

**Table 1 diagnostics-12-02686-t001:** Laboratory values for antibiotic treatment.

Laboratory	Admission	2 Weeks after Admission	Discharge	1-Month Follow-Up
WBC	5.4 × 10^9^/L	11.1 × 10^9^/L ↑	7.2 × 10^9^/L	6.5 × 10^9^/L
CRP	126.5 mg/L ↑↑	34.7 mg/L ↑	19.0 mg/L ↑	0.8 mg/L
AST,	48 U/L ↑	24 U/L	84 U/L ↑	35 U/L ↑
ALT,	39 U/L ↑	14 U/L	76 U/L ↑	23 U/L
GGT	46 U/L ↑	56 U/L ↑	678 U/L ↑↑	46 U/L ↑
D-dimer	5606 ug/L ↑↑	7174 ug/L ↑↑	4940 ug/L ↑↑	110 ug/L

Laboratory examination throughout the period of antibiotic treatment. Gradual recovery of laboratory values for aforementioned parameters. ↑—mildly increased level, ↑↑—highly increased level.

## Data Availability

Not applicable.

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
