# Peer review of "Aggressive Pyogenic Spondylitis Caused by S. constellatus: A Case Report"

_diagnostics, 2022, doi:10.3390/diagnostics12112686_

Round 1

Reviewer 1 Report

This paper concerns a case report of pyogenic spondylitis caused by Streptococcus constellatus. There was associated extensive abscess formation along the paravertebral muscles and intracranial gas formation. Treatment consisted of drainage and a 6-week course of antibiotics and the patient completely recovered.

I only have a few minor comments regarding this paper:

Page 1, Abstract, last line. "lumbar drainage" should preferable be changed to "abscess drainage from the lumbar zone". 

Page 1, Introduction, first line. Gram should be with a capital G.

Page 2, 1.2.Streptococcus constellatus. Fifth line from bottom of this paragraph. ..penicillin/ beta lactam. This is not clear. Probably the authors wished to write beta-lactam/beta-lactam inhibitor combination?

Page 4, first paragraph, first line. During an early post-operative period... The word "an" should preferable be changed to "the".

Page 4, first paragraph, 3rd line. ...and 2 g every 8 hours, respectively

Same page and paragraph, next to last line. ..excluded, should preferably be changed to discontinued.

Same page, second paragraph, 4th line. ...Consistent laboratory examinations etc. This sentence should be rephrased. ... for a (rather than the) total of thirty days...

Table 1: adds little information. Either add the real numbers, or consider replacing the table by only the relevant data in the text.

Page 5-7, Discussion. Is very detailed, partly repeats info from the Introduction, and can be abbreviated without losing relevance. 

Author Response

Dear Reviewer 1,

Thank You for Your constructive criticism and suggestions in order to improve our manuscript.

Point 1. Page 1, Abstract, last line. "lumbar drainage" should preferable be changed to "abscess drainage from the lumbar zone". 

Response 1. In lines 23-24 'lumbar drainage' was changed to 'abscess drainage from the lumbar zone'.

Point 2. Page 1, Introduction, first line. Gram should be with a capital G.

Response 2. In line 31 Gram is capitalized.

Point 3. Page 2, 1.2.Streptococcus constellatus. Fifth line from bottom of this paragraph. ..penicillin/ beta lactam. This is not clear. Probably the authors wished to write beta-lactam/beta-lactam inhibitor combination?

Response 3. In lines 78-79 Your suggestion has been written. 'Penicillin / beta-lactam' has been changed to 'beta-lactam / beta-lactam inhibitor combination'.

Point 4. Page 4, first paragraph, first line. During an early post-operative period... The word "an" should preferable be changed to "the".

Response 4. In line 120, 'an' was changed to 'the'.

Point 5. Page 4, first paragraph, 3rd line. ...and 2 g every 8 hours, respectively

Response 5. In line 122, the word 'respectively' has been added.

Point 6. Same page and paragraph, next to last line. ..excluded, should preferably be changed to discontinued.

Response 6. In lines 127-128 the word 'excluded' has been changed to 'discontinued'.

Point 7. Same page, second paragraph, 4th line. ...Consistent laboratory examinations etc. This sentence should be rephrased. ... for a (rather than the) total of thirty days...

Response 7. In line 137 the sentence is rephrased accordingly, 'the' was changed to 'a'.

Point 8. Table 1: adds little information. Either add the real numbers, or consider replacing the table by only the relevant data in the text.

Response 8. Table 1. in lines 150-152 was altered and now displays actual laboratory values.

Point 9. Page 5-7, Discussion. Is very detailed, partly repeats info from the Introduction, and can be abbreviated without losing relevance. 

Response 9. Discussion was altered. Lines 178-180 are shortened. The sentence in lines 183-185 is excluded. The sentence in lines 202-203 has also been excluded. Sentences in lines 211-214 have also been condensed. However, lines 224-231 have been added according to comments from the other Reviewer. Lines from 242-245 have been excluded. Sentences in lines 252-263 have been shortened.

Reviewer 2 Report

This is an interesting case report of a complicated  S. constellatus spondylitis involving paravertebral muscles and extra-spine sites (i.e. brain with pneumoencephalon and multiple gas inclusions). To my opinion, the manuscript shows a number of drawbacks and should be revised according to the following suggestions. Furthermore, english should be improved. 

1. First of all, the present diagnostic work-up did not include an echocardiogram: please, discuss this issue and explain your reasons.

2. lines 89-91: the 'radiological term' purulent is not acceptable, please use collection or similar term  

3. The past history of the patient includes an undefined cardiomyopathy: did she report details on this issue ? if yes, report them

4. Table 1 including increased or normal laboratory values is hazy as such for the reader; please, re-write it specifying absolute values or delete the table and report the lab values in the text

5. In the discussion section, please discuss these 2 papers relevant to this study:

POTSIOS C et al. Case Rep Infect Dis 2019; DOI 10.1155/2019/9364951

WANG TD et  al. Scan J Infect Dis 1996; DOI 10.3109/00365549609027179

6.   The conclusion section should be improved. Lower back pain and radiculopathy must be assessed by a physician as soon as possible. In addition, the diagnostic work-up should include radiologic evaluation and, if infectious spondylitis is suspected, imaging-guided or open surgery sampling for microbiological evaluation. Indeed, empirical antibiotic therapy should be avoided with the exception of some conditions such as  concomitant sepsis. To my opinion, the latter should be clearly underlined. Finally, the conclusions should include a proactive message specifically concerning the case described.

Author Response

Dear Reviewer 2,

Thank You for Your constructive criticism and suggestions in order to improve our manuscript.

Point 1. First of all, the present diagnostic work-up did not include an echocardiogram: please, discuss this issue and explain your reasons.

Response 1. Our patient underwent echocardiogram during the hospitalization with uneventful findings. Please see the added lines 129-131 which describe the findings.

Point 2.2. lines 89-91: the 'radiological term' purulent is not acceptable, please use collection or similar term

Response 2. In lines 89-91 'purulent compartment' was changed to 'collection'.

Point 3. 3. The past history of the patient includes an undefined cardiomyopathy: did she report details on this issue ? if yes, report them

Response 3. The patient did not elaborate about her cardiomyopathy which is explained in lines 96-98.

Point 4. Table 1 including increased or normal laboratory values is hazy as such for the reader; please, re-write it specifying absolute values or delete the table and report the lab values in the text

Response 4. Table 1. in lines 150-152 was altered and now displays actual laboratory values.

5. In the discussion section, please discuss these 2 papers relevant to this study:

POTSIOS C et al. Case Rep Infect Dis 2019; DOI 10.1155/2019/9364951

WANG TD et  al. Scan J Infect Dis 1996; DOI 10.3109/00365549609027179

Response 5. The paper by Wang et al. is now discussed in lines 224-228. The paper by Potsios et al. is discussed in lines 228-231.

Point 6.   The conclusion section should be improved. Lower back pain and radiculopathy must be assessed by a physician as soon as possible. In addition, the diagnostic work-up should include radiologic evaluation and, if infectious spondylitis is suspected, imaging-guided or open surgery sampling for microbiological evaluation. Indeed, empirical antibiotic therapy should be avoided with the exception of some conditions such as  concomitant sepsis. To my opinion, the latter should be clearly underlined. Finally, the conclusions should include a proactive message specifically concerning the case described.

Response 6. The conclusion section has been altered according to Your comments. We put an emphasis in prompt clinical evaluation by a physician, in addition to radiologic examination and imaging-guided or open surgery. Additionally, the comment about avoiding empirical antibiotic treatment is added in lines 277-279.

Round 2

Reviewer 2 Report

No further comment, the manuscript is fine and interesting for the readers od this Journal.